# Caring for a Child with Congenital Adrenal Hyperplasia Diagnosed by Newborn Screening: Parental Health-Related Quality of Life, Coping Patterns, and Needs

**DOI:** 10.3390/ijerph20054493

**Published:** 2023-03-03

**Authors:** Laura Rautmann, Stefanie Witt, Christoph Theiding, Birgit Odenwald, Uta Nennstiel-Ratzel, Helmuth-Günther Dörr, Julia Hannah Quitmann

**Affiliations:** 1Department of Medical Psychology, Center for Psychosocial Medicine, University Medical Center Hamburg-Eppendorf, Martinistraße 52, W26, 20246 Hamburg, Germany; 2Bavarian Screening Center, Bavarian State Office for Health and Food Safety, Veterinärstraße 2, 85764 Oberschleißheim, Germany; 3Department of Pediatric Endocrinology, University Children’s Hospital Erlangen, Loschgestr. 15, 91054 Erlangen, Germany

**Keywords:** congenital adrenal hyperplasia, 21-hydroxylase deficiency, newborn screening, psychological stress, parental needs, parental coping, rare disease

## Abstract

Diagnosing a child by newborn screening with classic congenital adrenal hyperplasia due to 21-hydroxylase deficiency (CAH) causes multiple challenges for the affected parents and the whole family. We aimed to examine the health-related Quality of Life (HrQoL), coping, and needs of parents caring for a child with CAH to develop demand-responsive interventions for improving the psychosocial situation of affected families. In a retrospective cross-sectional design, we assessed HrQoL, coping patterns, and the needs of parents caring for a CAH-diagnosed child using specific questionnaires. Data of 59 families with at least one child diagnosed with CAH were analyzed. The results show that mothers and fathers in this study reached significantly higher HrQoL scores compared to reference cohorts. Decisive for the above-average parental HrQoL were effective coping behaviors and the parental needs being met. These findings verify the importance of helpful coping patterns and rapid fulfillment of parental needs for maintaining a good and stable HrQoL of parents with a child diagnosed with CAH. It is crucial to strengthen the parental HrQoL to build a reasonable basis for a healthy upbringing and improve the medical care of CAH-diagnosed children.

## 1. Introduction

The introduction of expanded newborn screening in Germany in 2005, including rare congenital inborn errors of metabolism such as congenital adrenal hyperplasia (CAH), allows for diagnoses in newborns before the onset of symptoms. Therefore, early initiation of therapy can prevent newborns from severe consequences [1,2,3].

CAH is a rare disease with an incidence of 1:12,000 [3]. It is characterized by prenatal masculinization of the external genitalia in females. CAH may occur with or without a salt-wasting crisis. A salt-wasting crisis can lead to failure in thriving but also to shock or even coma. Children with CAH show an increased production of male hormones. When untreated, children will be short-statured. In adulthood, CAH patients often show obesity, metabolic changes, and infertility [4]. To prevent grave consequences such as potentially fatal salt-wasting crises or the above-mentioned long-term effects, it is crucial to detect CAH early in newborns. With an early diagnosis, it becomes possible to start the substitution of hormones rapidly and ensure a healthy growing up of the child. The newborn screening constitutes an efficient procedure to systematically screen for rare diseases with easy implementation by applying blood on filter paper.

CAH is characterized by an increased amount of 17-hydroxyprogesterone, which is crucial for the detection within the screening process. Approximately 94% of all CAH cases are allocated to classic CAH [5,6,7,8]. Classic CAH due to 21-hydroxylase deficiency (CAH) is an autosomal recessive disorder of cortisol biosynthesis caused by mutations in the active 21-hydroxylase gene (CYP21A2). In approximately two-thirds of patients with a 21-hydroxylase defect, there is a complete loss of function of the enzyme. This defect can lead to a life-threatening salt-wasting crisis within the first two to three weeks of life, which can recur in stressful situations without appropriate treatment. The clinical manifestations mainly include genital ambiguity in females due to androgen excess and life-threatening adrenal crises in both sexes due to impaired cortisol and aldosterone production [8]. In female children, external virilization of the genitals may occur, usually surgically corrected within the first two years of life [9]. In addition, despite treatment, patients usually achieve only below-average height and premature onset of puberty. The signs of hyperandrogenism may develop in girls, i.e., hirsutism, oligomenorrhea, and prolonging the menstrual cycle to more than 35 days [10].

Additionally, CAH is associated with many long-term implications relating to hormonal replacement therapy, clinical and biochemical monitoring, surgical interventions, fertility issues, and quality of life [5,6,7,8]. With a prevalence of approximately 1:14,000, CAH is a rare health condition [2].

Diagnosing a child with a chronic disease such as CAH causes multiple challenges for the parents and constitutes a profound change in parental thinking. The parents must understand and accept the disease and realize that the disease is lifelong. Many studies verify an increasing psychosocial burden on the entire family after a newborn is diagnosed with a chronic disease [11,12,13,14].

We know less about the psychosocial situation of parents of children with CAH, particularly those diagnosed by newborn screening. A few studies indicate an increasing strain on these parents, especially shortly after the initial diagnosis [6,15,16,17,18,19]. The initial shock about the diagnosis can lead to anxiety, grief, helplessness, feeling overwhelmed, and depressive symptoms [20].

Immediate and appropriate education by medical professionals about the diagnosis is essential in alleviating parents’ psychosocial distress. It is crucial that parents comprehensively understand the information [20,21,22]. Therefore, professionals must be trained in communication skills. In this conversation between parents and medical professionals, individual needs and gender-specific aspects must be considered. Fathers often need more medical information about their children’s health condition than mothers to gain sufficient understanding [23]. Parents associate uncertainty about the child’s health status with negative emotions. These negative emotions can negatively affect the parental relationship, especially in the case of genetically inherited conditions, affecting family dynamics [21]. As a result, there is a risk of disrupting the early attachment process to the child, which is of great importance for further child development. Accepting the diagnosis and adjusting expectations for one’s child is essential for developmentally supportive parenting behaviors [24]. Coping with the diagnosis depends on the family and social structures. However, it can be positively reinforced by specialized counseling, support services, and contact with other parents of CAH children [25,26].

Although there is evidence of increased psychosocial distress among affected parents, there is often a lack of professional support and psychosocial support services [27]. Due to the rarity and complexity of the diseases detected in newborn screening, there are usually problems finding specialized medical professionals to treat and care for the child [28].

In addition to the psychosocial burden, health-related quality of life (HrQoL) is an essential outcome of successful treatment. HrQoL describes a person’s subjective psychological, physical, mental, and social health-related well-being and functioning ability [29].

Studies examining the HrQoL of parents with chronically ill children reported a reduced HrQoL in parents caring for a chronically ill child due to an increased burden and additional responsibility [30]. Parents of children with chronic health conditions reported an increased time requirement, resulting in putting aside parental interests and social needs [27]. Due to the time-intensive necessary care of a chronically ill child, parents often reduce paid work, which may result in financial problems. In particular, mothers of chronically ill children often only work part-time or give up their employment as they are the primary caregivers [31]. Other factors include emotional distress, lack of support in housekeeping, or marital problems [30].

Only a few studies focus on HrQoL and the psychosocial consequences for the parents of children with CAH. Waldthausen [19] reported an average to above-average HrQoL of parents of CAH children compared to parents of healthy children. However, the parental burden seemed to depend on the children’s age and was highest in the postnatal period. Parents experienced feelings of overwhelming shock when receiving the child’s diagnosis. Due to insufficient information that is understandable for non-medical people, parents reported additional frustration and uncertainty [16]. Inadequate parent education includes missing information about the condition and the treatment (e.g., medication administration), coping strategies, and support in talking about the health condition with the affected child, siblings, family, and friends [32]. Parents of CAH children reported an increased need for psychosocial support; however, this support was often unavailable [33].

Nevertheless, parents seem to cope with their child’s chronic health condition over time and worry less about the child. 97% of parents of children with CAH aged 4 to 12 years reported being satisfied with their child’s health, and 76% had no fear of a salt-wasting crisis [34].

However, it is crucial to understand the needs of affected families to improve the psychosocial well-being of parents caring for a child with CAH. Gathering the dimensions of the HrQoL, coping patterns, and the specific needs of affected parents allows clinicians and researchers to gain insight into these families and thus provide adequate support. Therefore, our study aims to examine the HrQoL, coping patterns, and special needs of parents caring for a child with CAH.

## 2. Materials and Methods

This retrospective cross-sectional study examined the HrQoL, coping patterns, and needs of parents caring for a child with CAH diagnosed by newborn screening in Bavaria, Germany. It was a cooperation project of the Department of Medical Psychology of the University Medical Center Hamburg-Eppendorf (UKE), the Bavarian State Office for Health and Food Safety, and the University Children’s Hospital in Erlangen. In a mixed-method approach, quantitative and qualitative data were analyzed using validated questionnaires and semi-structured telephone interviews. The presented analysis focuses on the quantitative data, aiming to gain knowledge about HrQoL, coping patterns, and the needs of parents of children born with CAH. The Bavarian Medical Association Ethics Committee approved the study in September 2018 (No. 18003).

### 2.1. Participants

The Bavarian State Office for Health and Food Safety registry identified the families and asked them to participate in this study. Inclusion criteria were: (1) parents of children with classic CAH (21-OH deficiency), (2) children aged up to 18 years, and (3) CAH diagnosis by newborn screening in Bavaria [35,36]. The newborn screening is regularly performed 36 to 72 h after the child’s birth. If screening results are positive, parents were informed immediately for further examinations of their newborn.

Parents were excluded because of (1) missing informed consent, (2) insufficient knowledge of the German language, or (3) other diseases than CAH being the center of attention of the family.

### 2.2. Data Collection

The Bavarian State Office for Health and Food Safety sent questionnaires to eligible families via the postal service, including an informative letter about this study, a declaration of consent, and a pre-paid return envelope between September 2018 and September 2019. Parents reported sociodemographics, the child’s health condition, and circumstances regarding the diagnosis process. Based on self-reports, parents provided information about their HrQoL within the last week, coping strategies for dealing with the situation due to their child’s disease, and their current particular needs.

Parental HrQoL was measured using the chronic–generic Ulm Quality of Life Inventory for Parents (ULQIE) [37]. The ULQIE consists of 29 items divided into 5 subscales. The subscales include the dimensions (1) physical and daily functioning, (2) satisfaction with the situation in the family, (3) emotional distress, (4) self-development, and (5) well-being, as well as four single items without any scale assignment. Each item is rated on a 5-point Likert scale from 0 to 4. The scales’ internal consistency (Cronbach’s Alpha) varies between 0.74 and 0.92.

Parental coping strategies were assessed using the chronic–generic Coping Health Inventory for Parents (CHIP) questionnaire [38,39]. The CHIP consists of 45 items that examine the usefulness of different coping strategies of parents of chronically ill children. The items are assigned to three subscales: (1) maintaining family integration, cooperation, and an optimistic definition of the situation; (2) maintaining social support, self-esteem, and psychological stability; and (3) understanding the medical situation through communication with other parents and consultation with medical staff. All items are rated on a 4-point Likert scale from zero to three. Cronbach’s Alpha for the scales is between 0.79 and 0.90 [39].

Chronic–generic needs of parents were measured using the Scale of needs for parents of chronically ill children. This questionnaire contains 19 items about the parental need for information, psychosocial care, exchange with others, and support in everyday life. The items are rated on a 5-point Likert scale ranging from 1 to 5: the higher the item’s score, the more intense the parent’s need. The reliability of the Scale of needs for parents of chronically ill children shows internal consistency of 0.95 [40].

### 2.3. Data Analysis

We used the statistic software SPSS, version 25 (IBM, Armonk, NY, USA, 2017) for data analysis. Descriptive statistics were computed for each subscale of the questionnaires examining HrQoL and coping. Missing values were replaced by the individual mean score for each variable if missing data were random or less than 20.00% of all scale items [41]. We compared the scores from our sample to reference values using Student’s *t*-tests. Multiple linear regression analyses were performed to identify variables that predict parental HrQoL. Sociodemographic and clinical variables, the CHIP total score, and the total score of the Scale of needs for parents of chronically ill children were entered as predictors into the model. The clinical data comprised the child’s age, sex, and current hormonal replacement therapy. Since no data on the pubertal status of the children were reported, we set the age of 10 years, according to Sawyer, Azzopardi [42], as the beginning of adolescence. If families live with more than one CAH-diagnosed child, we used the data for the firstborn CAH child for further analysis.

The examined sociodemographic aspects consisted of the parents’ age and sex, number of children, number of children diagnosed with CAH, place of residence, educational level, and employment status in the last 12 months. Further examined aspects were the use of psychosocial consultation and contact with other parents who have children with CAH.

For the confirmatory approach, a Bonferroni adjustment for multiple testing was conducted. Based on an initial significance level of α = 0.05, the significance level was reduced to α = 0.0033. The reduction was justified by simultaneously calculating multiple linear regression analyses and one one-sample *t*-test. The linear regression model investigates 14 relevant influencing factors to analyze the influence on the HrQoL measured by the ULQIE total value. With the one-sample *t*-test, the computed ULQIE total value was compared to a reference cohort’s total value.

Exploratory regression analyses were conducted to investigate the impact of clinical or sociodemographic aspects on the CHIP’s total score, the total score of the Scale of needs for parents of chronically ill children, and each subscale of the ULQIE. In addition, we explored whether the coping behavior or parents’ intensity of needs influenced the ULQIE-subscales.

## 3. Results

The 120 eligible families received the study information as well as the questionnaires. Of these families, 62 agreed to participate and filled out the questionnaires, resulting in a response rate of 51.67%. Among the 62 families with at least one child diagnosed with CAH, three were excluded due to one or more exclusion criteria, resulting in a final sample size of 59 families (Figure 1).

### 3.1. Sample Characteristics

We included 59 families in the final analysis, consisting of 100 parents (59 mothers and 41 fathers). The mean age of the parents was 43.18 ± 7.48 (SD) years, with 42.12 ± 7.50 for mothers and 44.71 ± 7.27 (SD) for fathers. In 41 families, both parents answered the questionnaires, whereas in 18 families, only the mother participated. The mean time of conducting the presumptive diagnosis of the child was 7.79 days after birth. Ten of the fifty-nine families had two affected children diagnosed with CAH, including two families giving birth to twins with CAH (Figure 1). The number of children living in the household ranged between 1 and 5, with a mean of 2.13 ± 0.91 (SD). Fifty-nine children with CAH (26 girls, 33 boys) plus ten siblings diagnosed with CAH (six girls and four boys) were identified. The mean age of the children was 11.01 ± 4.86 (SD) years (range: 0.68–18.12 years). Parents of 23/32 girls reported virilization of external genitalia at birth, and genital surgery was already performed in 22 girls. All children were treated with hydrocortisone, and 56 children (81.16%) received fludrocortisone. Further detailed information concerning parents’ and children’s characteristics and psychological support at the time of diagnosis are listed in Table 1, Table 2 and Table 3.

### 3.2. HrQoL of Parents

Parental HrQoL scored in the upper segment of the 0–4 scale, with the highest means for the subscale satisfaction with the situation in the family (3.38 ± 0.59) and the lowest means for the subscale self-development (2.28 ± 0.77) (Table 4). In comparison to the used reference values [37], parents of CAH children reported significantly higher HrQoL scores for physical and daily functioning (*t*(97) = 9.15, *p* < 0.01), emotional distress (*t*(97) = 12.31, *p* < 0.01), self-development (*t*(97) = 4.53, *p* < 0.01), well-being (*t*(98) = 6.53, *p* < 0.01), as well as for the ULQUIE total score (*t*(97) = 8.36, *p* < 0.01).

No statistically significant differences between the HrQoL scores of mothers and fathers were found. The same results were found for parents of one affected child with CAH compared to parents of two CAH children. However, the analyses showed differences in the ULQUIE total score between parents of younger (<10 years) and older CAH children (≥10 years), as well as between parents of CAH girls and CAH boys (Table 5). Parents of children with CAH in adolescence reached higher ULQIE total scores than parents of children younger than ten years. Further, parents of a girl with CAH seemed to have a higher HrQoL than parents of a CAH boy.

The regression analysis explained a substantial variance in parental HrQoL assessed using the ULQIE total score (F_(14,44)_ = 1.83, *p* = 0.06, *n* = 58). The results showed that sociodemographic or clinical factors did not significantly affect the total score of the ULQIE. Nonetheless, the examined HrQoL of the parents depended considerably on the effectiveness of applied coping strategies and the intensity of the needs of these parents. Confirmatory regression analyses showed a significant positive correlation between the total score of the CHIP and the ULQIE total value. Additionally, the data conveyed that parents with intense needs achieved significantly lower HrQoL scores (Table 6). The coefficient of determination R^2^ for the linear regression model was 0.40.

Exploratory regression analyses showed a correlation between particular subscales of the ULQIE and several factors involved. The subscales daily functioning, family satisfaction, and emotional distress seemed to depend on coping strategies and needs in the same way as the total score of the ULQIE (Table 6). The HrQoL measured by the subscale family satisfaction was further influenced by the child’s age (r = −0.05, *p* = 0.01) and the use of psychosocial consultation (r = −0.49, *p* = 0.04). Parents who used psychological or social-pedagogic help reported a lower HrQoL, measured by the family satisfaction subscale. The same is true for parents with an older CAH child. Concerning the subscale self-development, regression analyses showed impacts from the parents’ educational qualification, the gender of their child, and their applied coping strategies. The higher the parental educational degree, the lower the HrQoL measured by the subscale self-development (r = −0.48, *p* = 0.04).

Additionally, the HrQoL on the self-development scale of parents of a boy decreased by about 0.45 units compared to parents of a girl (*p* = 0.06). The coping strategies influenced the parents’ self-development in the same way as the total score (Table 7). The subscale well-being resulted in high *p*-values. Therefore, no well-founded tendencies could be suggested. Nevertheless, we did not prove these findings in a significant manner due to the exploratory approach.

### 3.3. Coping of Parents with CAH Children

The parents’ individual coping behavior was measured using the CHIP. Mothers and fathers in this study stated that maintaining family integration, cooperation, and an optimistic definition of the situation was their most helpful coping pattern (2.31 ± 0.45). Maintaining social support, self-esteem, and psychological stability was deemed less helpful than the other subscales. With a mean total score of 2.01 (±0.47), the effectiveness of coping mechanisms of parents of children with CAH was comparable to German reference cohorts [39,44]. Mothers rated the helpfulness of particular coping patterns higher than fathers (Table 8).

The exploratory regression analyses showed high *p*-values (*p* > 0.11) on every independent variable, including sociodemographic or clinical particulars for the CHIP total score as the dependent variable within the model.

### 3.4. Needs of Parents with CAH Children

We measured the intensity of the needs of parents with CAH-diagnosed children using the Scale of needs for parents of chronically ill children. Comparing mothers and fathers, the mean total scores differed by about 0.12 points, resulting from mothers’ mean score of 2.87 (±0.83) and a mean score of 2.75 (±0.76) for fathers. The most frequently indicated needs were primarily information, such as needing more information about therapy options, dealing with departments and councils, diagnostic methods, and dealing with the ill child. Another often-mentioned requirement was having more time for the partner. The three needs least indicated were the need to obtain relief in everyday life from relatives or friends and to conduct problem-centered conversations with friends (Table 9).

Furthermore, parent-reported needs showed differences depending on the child’s gender. Exploratory regression analyses indicated that parents of a boy declared their needs more severely than parents giving birth to a girl with a 0.69 unit difference (*p* ≤ 0.01). Other factors included in the model did not show any substantial dependences.

## 4. Discussion

Diagnosing a child with CAH raises multiple challenges for the parents concerning lifelong hormonal substitution therapy, potential risk of adrenal crisis, feminizing surgery, gender identity, and long-term morbidity. From the start of the diagnosis, the parents relied on healthcare providers, particularly the pediatric endocrinologist, to provide them with all information about the disease and their child’s care [16]. Conflicting recommendations by physicians, nurses, and persons in the parental environment can cause parental uncertainties [32]. A few studies have pointed out an increasing psychological burden on these parents [6,15,16,18,19,45]. However, overall data on the psychological adjustment of parents are scarce. This study investigated parental HrQoL, coping strategies, and needs in families with CAH children.

The parents of CAH children in our study reported an above-average HrQoL compared to data from other pediatric samples of oncology patients, patients with a cardiological disease, or patients with cystic fibrosis [43,46]. We assume the time since diagnosis might be decisive for the parental HrQoL. At the time of the study, the mean age of the CAH children diagnosed by newborn screening was 11.01 years, whereas the cited study was performed during the first three months after diagnosis [47]. It has been shown that HrQoL increased over time [43].

Our results show that the most important influencing factors on the HrQoL of parents of children with CAH are the effectiveness of applied coping patterns and the intensity of needs among these parents. Regression analyses revealed that effective coping and low intensity of particular needs positively affect parental HrQoL, and no other influencing factors significantly impact the parental HrQoL. Although the R^2^ marks 0.40, this regression model sufficiently explains the influence on the parental HrQoL since we included many possible influencing factors. The parents in the present study rated maintaining family integration, cooperation, and an optimistic definition of the situation as their most helpful coping patterns. Moreover, the surveyed mothers and fathers reached the highest means of the HrQoL measurement at the subdimension satisfaction with the situation in the family, which emphasizes the importance of well-functioning family structures. This observation is congruent with Van Schoors et al. [48], who found that the parents reported a better HrQoL if they experienced higher emotional closeness in their families. In particular, the perceived level of expressiveness within the family was shown to be decisive [48].

McCubbin et al. [39], Senger et al. [49], and Clever et al. [44] showed that the subdimension maintaining family integration, cooperation, and an optimistic definition of the situation was evaluated as most helpful by parents. These results underline the potential of enhancing family cohesion as a critical coping pattern to improve parental HrQoL; the child’s specific disease might not be as important.

Additional important coping patterns influencing the parental HrQoL are understanding the disease itself and a feeling of competence in disease management. Disturbed communication with healthcare providers, especially during diagnosis and subsequent treatment guidelines, negatively impacts on parental management of the disease [16]. A limited understanding of CAH, especially its genetic implications, including recurrence risk and carrier status, was detected in a small study of five CAH families from Manila [50]. On the other hand, Fleming, Van Riper [51] showed a significant, positive relationship between detailed provider instruction on managing adrenal crises and perceived parental management ability in parents of children with CAH. They reported that the impact of CAH on the family decreases with a gain in parental management ability. A direct link between perceived management ability and parental HrQoL was found in children with cancer [48]. These findings highlight the importance of understanding and competence in managing upcoming disease-related challenges for good parental HrQoL.

With the first three needs most frequently reported being explicitly informational, parents pointed to receiving sufficient information about their child’s disease and upcoming management challenges. The findings are comparable to the results provided by parents of children with other chronic diseases [40]. As elaborated earlier, well-informed parents have a higher perceived management ability and reach higher HrQoL scores [17,48]. Therefore, it is crucial to convey thorough and understandable information and, in this way, meet parental needs as early as possible.

In the context of information collection, parents in the present study accentuated contact with other affected families. Almost one-third stated initiating active contact with other parents in the same situation through disease-specific patient self-help groups. Furthermore, a detailed information educational brochure about CAH presented by a patient organization was appreciated as very supportive. The parents described the patient organization as helpful for gaining information, practical advice, and social support. However, around half of our sample’s parents (59.57%) did not contact other affected families. Parent-to-parent communication is a meaningful intervention to improve the knowledge, self-efficacy, and HrQoL of mothers of ill children [52]. Other studies have highlighted self-help groups’ positive effect on patients’ HrQoL [53,54]. These findings suggest that parent-to-parent contact for mothers and fathers of CAH children is desirable and highlight the need for early involvement in patient-organizations, in the best case, parallel with the initial diagnosis.

Parents in this study did not declare a high need for psychosocial support at the survey time. Altogether, only 16.00% of the mothers and fathers used psychosocial consultation in the past. However, about two-thirds of this group evaluated psychosocial support as helpful or very helpful. Studies have shown that psychological support, e.g., through cognitive behavioral therapy, is highly recommended for parents of children with chronic health issues. Multiple emotional stressors and coping challenges for the families of children with CAH can increase the risk of developing psychopathological symptoms [55]. Immense psychological stress of the parents was reported in a study of parents in Sri Lanka [56].

The clinical practice guideline published by the Endocrine Society in 2010 also recommends regular behavioral/mental health consultation and evaluation for parents to address any concerns related to CAH [5].

We assume that the parents’ need for psychosocial help was no longer present at the time of this survey since, in most cases, many years have passed after disclosing the child’s diagnosis. Another reason could be that there was impeded access to psychosocial consultation. Approximately only every tenth family in the present study received a professional offer of psychosocial support during their first clinical stay after learning of the presumptive diagnosis of their child. Consequently, early psychosocial support could be a valuable topic with scope for improvement.

## 5. Practical Recommendations

Different dimensions of care should be considered to improve the situation of CAH-affected families. Sensitizing physicians at maternity and pediatric clinics to this rare disease is essential. Medical professionals should be encouraged to consult experts early, such as pediatric endocrinologists, to accompany respective families from the diagnosis process onwards. In addition, our results show that even after a mean time of 11 years after the disclosure of the CAH diagnosis of their child, mothers, and fathers still indicated more information regarding the disease as being their primary need. This knowledge gap implies that with a complex disease such as CAH, new arrival questions, and informational needs are shared continuously.

Consequently, improving the mediation of first and ongoing information is essential for improving care for these families. Another critical aspect constitutes the mediation of the (presumptive) diagnosis. Studies show that parents value thorough, timely, and understandable information without too many specialist terms [16,18]. Furthermore, physicians should be aware of the process of announcing the diagnosis of a child as a potentially traumatizing event for mothers and fathers [16]. Therefore, it is recommendable to additionally offer: (1) written material with accessible and understandable information about the disease, upcoming obstacles, and practical advice, as well as (2) early psychosocial support for the entire family directly at the disclosure of the diagnosis.

Furthermore, families should be encouraged to participate in self-help groups early. With the results of this study, the development of demand-responsive, purposeful interventions can be initiated. Family-oriented coping should focus on early interventions to strengthen family solidarity and improve communication. Another leading aspect depicts information-oriented coping. Early and repeated training for managing adrenal crises is desirable to strengthen the mother’s and father’s management ability and improve the situation of CAH-affected families.

## 6. Conclusions

A child with a chronic disease diagnosis marks an unexpected, extraordinary, and stressful change in the prevailing family-life system. Since CAH constitutes a rare chronic disease with a potentially life-threatening character, parents must overcome additional obstacles in caring for their children. Thus, they are even more at risk of undergoing psychological strain and decreasing their HrQoL [56,57,58]. Parental well-being is fundamental for the child’s psychosocial development [12,59]. Therefore, it is essential to consider the psychosocial situation of parents of children diagnosed with CAH. The present study is the first to examine parental HrQoL in families with pediatric CAH patients extensively. The results show that despite the complex and seemingly threatening nature of CAH, parents are doing pretty well and do not experience extensive restrictions in their daily lives through their child’s disease. The present data imply an above-average HrQoL in mothers and fathers of children and adolescents with CAH.

On the one hand, the established parental HrQoL is highly affected by the effectiveness of the mother’s and father’s applied coping patterns. Parents in the present study described their coping strategies as helpful so that effective coping can be assumed. On the other hand, the intensity of needs in parents of a child with CAH significantly influences their perceived HrQoL. Hence, this study verifies the importance of helpful coping patterns and the rapid fulfillment of parental needs for maintaining a good and stable HrQoL of parents with a child diagnosed with CAH.

## 7. Strengths and Limitations

To our knowledge, this is the first study investigating the HrQoL of parents of CAH-diagnosed children. The findings provide important impulses for improving the situation of CAH-affected families since decisive influencing factors on the parental HrQoL were found.

Even though the examinations concern a rare disease, many participants could be gathered for this study. Questionnaires were addressed to 83.90% of all registered parents with a child diagnosed with CAH through newborn screening in Bavaria. With a response rate of 51.67%, the sample covers a large number of affected families in this region. Further, contrary to many other studies, mothers and fathers were included in the survey to include the family perspective. A strength of this study is based on the applied methods. Using a Bonferroni adjustment for multiple testing ensures the prevention of possible interference factors to a large extent.

Nevertheless, various limitations of the study have to be acknowledged. Since the study format was retrospective and not based on clinical data, recall bias cannot be excluded. Further, even though there was a satisfying number of returns, the parents who did not participate may have struggled with their situation the most and could not answer the survey. An important fact to add is that the children’s ages in this study were spread over a wide range (0.68–18.12 years). Therefore, parents newly confronted with their child’s initial diagnosis and parents who had years to adapt to the new situation filled out the questionnaires, making the study population heterogeneous. The present study also describes a regional sample, including only parents in Bavaria, Germany. However, this study still delivers a profound gain in knowledge about the psychosocial situation of CAH-affected families and elaborates important impulses to improve the provision of care. Further studies should concentrate on accompanying families with CAH-diagnosed children early on and in other federal states and countries. A longitudinal format is suggested.

## Figures and Tables

**Figure 1 ijerph-20-04493-f001:**
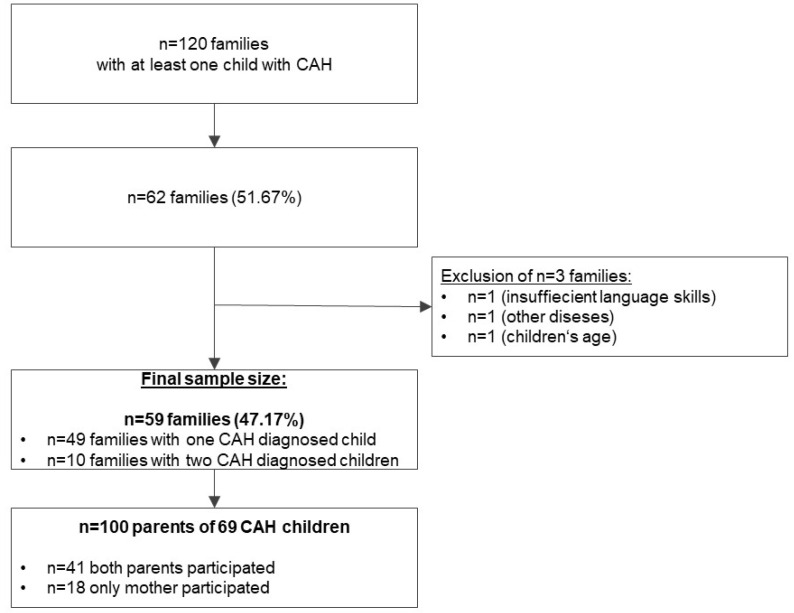
Flowchart of the recruiting process.

**Table 1 ijerph-20-04493-t001:** Data of Parents’ Characteristics (*n* = 100).

		*n*	(%)
Gender (*n* = 100)	female	59	(59.00)
	male	41	(41.00)
Marital status (*n* = 97)	single parent	6	(6.00)
	partnership	91	(91.00)
Place of residence (*n* = 99)	large town ^a^	11	(11.00)
	middle town ^b^	19	(19.00)
	small town ^c^	16	(16.00)
	rural area ^d^	53	(53.00)
Educational qualifications (*n* = 100)	low ^e^	30	(30.00)
	middle ^f^	38	(38.00)
	high ^g^	32	(32.00)
Employment (*n* = 99)	full-time job	49	(49.00)
	part-time job	35	(5.00)
	parental leave	7	(7.00)
	other ^h^	8	(8.00)

Notes: *n* = numbers ^a^ ≥100.000 residents; ^b^ ≥20.000–99.999 residents; ^c^ ≥5.000–19.999 residents; ^d^ <5.000 residents; ^e^ no qualification or certificate of secondary education; ^f^ ordinary level or A-level; ^g^ bachelor’s or master’s degree; ^h^ self-employment, housewife, mini-job.

**Table 2 ijerph-20-04493-t002:** Data of Childrens’ Characteristics (*n* = 69).

		*n*	(%)
Gender (*n* = 69)	female	32	(46.38)
	male	37	(54.62)
Virilization of outer genitals (only females) (*n* = 32)	yes	23	(71.88)
	no	9	(28.12)
Admission to hospital after the first year of life because of salt-wasting crisis/low blood sugar level (*n* = 68)	no	41	(60.29)
	yes, once	13	(19.12)
	yes, repeatedly	14	(20.59)
		**M**	**(SD)**
Age of firstborn CAH-children (*n* = 59)		11.14	(12.32)
Age of later born CAH-children (*n* = 10)		10.25	(3.63)
Age of the child at presumptive diagnosis (in days) ^a^ (*n* = 59)		7.79	(6.74)

Notes: *n* = numbers; M = mean; SD = standard deviation; ^a^ if more than one child with CAH is living in the family, the data of the firstborn CAH-child are reported.

**Table 3 ijerph-20-04493-t003:** Use of psychosocial support.

		*n*	(%)
Use of psychosocial consultation due to CAH-specific burden (*n* = 100)	no	84	(84.00)
	yes	16	(16.00)
Was the psychosocial consultation helpful? (*n* = 16)	not at all	2	(12.50)
	a little/partly	4	(25.00)
	much/a lot	10	(62.50)
Contact with other affected parents ^a^ (*n* = 94)	self-help group	30	(31.91)
	internet forums	8	(8.51)
	social networks	6	(6.38)
	other ^b^	5	(5.32)
	none	56	(59.57)
Any offer of psychosocial ^a, c^ support at the ward (after birth, initial diagnosis) (*n* = 83)	no	75	(90.36)
	yes	8	(9.64)

Notes: *n* = numbers, ^a^ multiple responses possible; ^b^ telephone, private contacts; ^c^ psychologist, social workers, other than medical staff.

**Table 4 ijerph-20-04493-t004:** HrQoL of parents (ULQIE)—Comparison with Reference Data.

	Domains	*n*	Mean (SD)	Ref	t	df	*p*
HrQoL	Physical and daily functioning	98	3.01 (0.62)	2.44a	9.15	97	<0.01
	Family satisfaction	96	3.38 (0.59)	3.28a	1.65	95	0.10
	Emotional distress	98	3.12 (0.66)	2.30a	12.31	97	<0.01
	Self-development	98	2.28 (0.77)	1.93a	4.53	97	<0.01
	Well-being	99	3.12 (0.67)	2.68a	6.53	98	<0.01
	Total	98	2.99 (0.51)	2.56a	8.36	97	<0.01
Percentile data	Mothers	55	77 (23.70)	55b	6.77	54	<0.01
	Fathers	40	80 (18.70)	60b	6.61	39	<0.01
	Total	95	78 (21.67)	70c	3.76	94	<0.01

Notes: df = degrees of freedom; HrQoL = health-related quality of life; *n* = numbers; *p* = *p*-value; Ref = reference value; SD = standard deviation; t = t-value; ULQIE = Ulm Quality of Life Inventory for Parents (Goldbeck and Storck, 2002); a with reference to Goldbeck and Storck [37], confirmatory approach; b with reference to West, Besier [43], exploratory approach; c with reference to Waldthausen [19], exploratory approach.

**Table 5 ijerph-20-04493-t005:** Parental Quality of Life (ULQIE total score)—Subgroup Analysis.

Subgroups	*n*	Mean	SD
Mothers	57	2.98	0.50
Fathers	41	3.00	0.53
Parents of children < 10 years	39	2.95	0.68
Parents of children ≥ 10 years	59	3.02	0.44
Parents of girls	41	3.07	0.53
Parents of boys	57	2.93	0.49
Parents: 1 CAH child	81	2.99	0.50
Parents: 2 CAH children	17	2.98	0.55

Notes: CAH = congenital adrenal hyperplasia; *n* = numbers; SD = standard deviation; ULQIE = Ulm Quality of Life Inventory for Parents (Goldbeck and Storck, 2002).

**Table 6 ijerph-20-04493-t006:** Influencing Factors on parental Quality of Life (ULQIE total score).

Influencing Factors	r	SE	*p*	95% CI
Constant term	2.20	0.52	<0.01	1.15–3.26
Gender of parent (mother)	−0.04	0.18	0.83	−0.39–0.32
Parental Age	0.02	0.01	0.14	−0.01–0.04
Number of all children	−0.01	0.07	0.85	−0.16–0.13
Number of CAH children	−0.10	0.16	0.55	−0.42–0.23
Place of residence (countryside)	0.05	0.12	0.69	−0.19–0.28
Educational qualification (high)	−0.01	0.13	0.95	−0.27–0.25
Employment (part-time job)	0.08	0.17	0.65	−0.27–0.42
Psychosocial consultation (yes)	−0.04	0.19	0.84	−0.42–0.34
Contact with other parents (yes)	−0.13	0.12	0.28	−0.38–0.10
Gender of the child (girl)	0.04	0.13	0.77	−0.22–0.30
Age of the child	−0.01	0.02	0.90	−0.03–0.03
Phenotype of CAH (SW)	−0.17	0.15	0.29	−0.48–0.15
CHIP total score	0.56	0.14	<0.01	0.27–0.84
Scale of parental needs	−0.27	0.08	<0.01	−0.43–0.11

Notes: CAH = congenital adrenal hyperplasia; CHIP = Coping Health Inventory for Parents; CI = confidence interval; r = coefficient of regression; *p* = level of significance; SE = standard error; SW = salt-wasting type; ULQIE = Ulm Quality of Life Inventory for Parents.

**Table 7 ijerph-20-04493-t007:** Influencing Factors on parental Quality of Life (ULQIE Subscales).

ULQIE Subscales	Daily Functioning		Family Satisfaction		Emotional Distress		Self-Development		Well-Being	
	r	*p*	r	*p*	r	*p*	r	*p*	r	*p*
CHIP ^a^	0.44	0.03	0.73	<0.01	0.39	0.08	0.83	<0.01	0.39	0.08
needs	−0.24	0.03	−0.32	<0.01	−0.35	0.01	−0.21	0.16	−0.16	0.20

Notes: CHIP = Coping Health Inventory for Parents; *p* = level of significance; r = coefficient of regression; ULQIE = Ulm Quality of Life Inventory for Parents, ^a^ total score.

**Table 8 ijerph-20-04493-t008:** Parental Coping (CHIP).

CHIP Domain		*n*	M	SD
Subscales	(1) Family integration	93	2.31	0.45
	(2) Social support	93	1.75	0.59
	(3) Medical knowledge	95	1.93	0.69
Total Score	both parents	92	2.01	0.47
	mothers	53	2.10	0.41
	fathers	39	1.90	0.52

Notes: CHIP = Coping Health Inventory for Parents; M = arithmetic mean; SD = standard deviation.

**Table 9 ijerph-20-04493-t009:** Parental Needs (Scale of needs for parents of chronically ill children).

Perspective	Items	*n*	Mean	SD
Scale of needs for parents of chronically ill children	Information about therapeutic options	96	4.02	1.13
	Information about dealing with departments and councils	91	3.47	1.40
	Information about diagnostic methods	88	3.45	1.38
	Having more time for my partner	86	3.44	1.06
	Information about dealing with my ill child	91	3.37	1.42
	Having more time for oneself	88	3.07	1.17
	Speaking more about my problems with my partner	82	3.05	1.41
	Meeting up with other affected parents	87	2.90	1.36
	Having more time for other family members	87	2.86	1.16
	Having more time for my other child(ren)	69	2.67	1.54
	Having more time for my friends	87	2.66	1.04
	Information about dealing with my other child(ren)	75	2.61	1.48
	Speaking more about my problems with other affected parents	85	2.56	1.27
	Getting more relief in everyday life from my partner	82	2.52	1.24
	Speaking more about my problems with relatives	85	2.48	1.25
	Having a contact in the treatment center to speak with about my problems (psychologist/social education worker)	86	2.34	1.23
	Getting more relief in everyday life from relatives	83	2.19	1.09
	Speaking more about my problems with friends	87	2.18	1.05
	Getting more relief in everyday life from friends	83	1.81	0.97
	Total	80	2.82	0.80

Notes: *n* = numbers; SD = standard deviation.

## Data Availability

The data supporting this study’s findings are available from the corresponding author upon reasonable request.

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
