# Peer review of "Caring for a Child with Congenital Adrenal Hyperplasia Diagnosed by Newborn Screening: Parental Health-Related Quality of Life, Coping Patterns, and Needs"

_ijerph, 2023, doi:10.3390/ijerph20054493_

Round 1
Reviewer 1 Report
The manuscript by Rautmann et al is interesting, well written, and fills worldwide data with new information about the psychological issues of parents with a child diagnosed with CAH. I have the few following comments:
Introduction: I recommend to fill information about when NBS for CAH was introduced and the incidence rate of classic CAH per year in Germany.
Line 40: Firstly, I recommend starting with a general description of CAH, then describing that 21-OHD is the most prevalent cause of the disease and how could be diagnosed 21-OHD (by the increased level of 17-OHP).
Line 52-52: I recommend writing that “The signs of hyperandrogenism may develop for girls, i.e. hirsutism, oligomenorrhea, prolonging the menstrual cycle to more than 35 days.”
Methods:
Line 179: “hormone replacement therapy” is used more commonly than “hormonal substitution therapy”
Results: The general part of the description of the patients’ cohort group could be filled with information about CAH forms (salt-wasting and simple virilizing), distribution of external genitalia virilization by Prader scale, and how it affected parents psychologically. Would be more clear if would describe 1 case from 23 with virilization, why was not apply feminizing surgery, was not applied due to age (too young yet according to protocol), or other reasons (etc. avoid parents).
Additional parts:
Line 476 and 479: Please correct the space between symbols.
Author Response
Dear Reviewer, thank you for your fruitful and concrete feedback, which helps us to improve the manuscript. We marked all changes in yellow.
Introduction: We added information about the starting point of NBS for CAH in Germany (line 36) and revised the introduction section adding detailed information about incidences and a general description of CAH (lines 42-49). We appreciated your suggestion for rewording line 52 and revised the sentence accordingly to your suggestion (lines 67-69).
Methods: We replaced "hormonal substitution therapy" using hormone replacement therapy" (lines 197-199).
Results: Thank you for recommending adding more information about the children's characteristics. Due to parental self-reports only, we cannot report further clinical data. Nevertheless, we added a table (table 2) to report children's characteristics in more detail (lines 231-232, lines 235-236, line 245).
Additional parts: We corrected the space between the symbols (lines 500 and 503).

Reviewer 2 Report
Rautmann et al describe the parental impact of the CAH diagnosis by newborn screening. The study is interesting and well structured, results are appropriate.
I have some minor consideration:
- in the introduction, the authors sould better explain the advantages of the NBS and early treatment on the disease course
- Authors declare that "the parental burden seemed to depend on the children age and was highest in the postnatal period. In the results they distinguish only parents of patients < 10 yrs or > 10 yrs. The range is very wide, more subgroups could better explain if there are differences based on children age.
- in the methods more information on the CAH screening are required, for example when it is performed and when parents are informed with the suspect/diagnosis
- pratical recommendations and strenght and limitation should be placed before the conclusions
Author Response
Dear Reviewer, we appreciate your feedback and suggestions for improving our manuscript on the quality of life and needs of parents of children diagnosed with CAH. We marked all changes in yellow.
Introduction: We added a section about the advantages of NBS and the early treatment (lines 49-54).
Methods: we added general information about the conduction of the NBS (lines 153-155) in the methods section and added a sentence in the results section describing the time when parents were informed about the presumptive diagnosis of CAH (lines 231-232).
Results/discussion: In the results section, we reported the parental HrQoL-Score and distinguished between children younger than ten years and children aged ten years and older. Due to the small sample size, we cannot subdivide into more age groups. The reported analyses are explorative and indicate that it is necessary to consider the child's age when assessing parental HrQoL, respectively, when researching about potentially influencing factors.
We placed the "practical recommendations " section and "strengths and limitations "before the conclusions.
